# Comprehensive Model for Evaluating the Performance of Mach-Zehnder-Based Silicon Photonic Switch Fabrics in Large Scale

**Marouan Kouissi [1], Benoit Charbonnier [1,\*] and Catherine Algani [2]**

[1]   Minatec Campus, University Grenoble Alpes and CEA-LETI, 38054 Grenoble CEDEX, France;
      marouan.kouissi@cea.fr

[2]   Esycom, University Gustave Eiffel, CNRS, CNAM, 75003 Paris, France; catherine.algani@lecnam.net

**\***   Correspondence: benoit.charbonnier@cea.fr

**Abstract:** Building a large-scale Mach-Zehnder-based silicon photonic switch circuit (LS-MZS) requires an appropriate choice of architecture. In this work, we propose, for the first time to our knowledge, a single metric that can be used to compare different topologies. We propose an accurate analytical model of the signal-to-crosstalk ratio (SCR) that highlights the performance limitations of the main building blocks: Mach-Zehnder interferometers (MZI) and waveguide crossings. It is based on the cumulative crosstalk and total insertion loss of the LS-MZS. Four different architectures: Beneš, dilated Beneš, switch and select, double-layer network were studied for the reason that they are mainly referenced in the literature. We compared them using our developed SCR indicator. With reference to the state-of-the-art technology, the analysis of the four architectures using SCR showed that, on a large scale, a high number of waveguide crossings significantly affects the performance of the switch matrix. Moreover, better performance was reached using the double-layer-network architecture. Then, we presented a 2 × 2 MZI using two electro-optic phase shifters and a waveguide crossing realized in LETI's silicon photonics technology. Measured performances were quite good: the switch circuit had a crosstalk of −31.3 dB and an insertion loss estimated to be less than 1.31 dB.

**Keywords:** signal-to-crosstalk ratio; switch architectures; Mach-Zehnder-based silicon photonic switch circuit; waveguide crossing

## 1. Introduction

Data switching networks are facing increasing difficulties in handling the exponential growth of traffic [1]. The scaling up of traditional network-on-chip capacity is severely limited by overheating problems [2]. Silicon photonics is believed to be a potential solution to improve the performance of interconnects and computing systems, specifically allowing larger bandwidth and lower power consumption. Data switching networks can thus benefit from several advantages of Si-technology such as large-area wafers, CMOS-compatibility, large-scale dimension device integration, high-level integration process and device density, heterogeneous integration such as electro-optic modulators and germanium photodetectors, low-cost, high volume, and compactness. Planar photonic switches have been largely fabricated on Silicon platforms, based on thermo-optic (TO) or/and electro-optic (EO) phase shifters. The first ones have a micro-second switching transient time, which matches the requirements of telecommunication network nodes and inter-data center connections. In contrast, the second ones have a nano-second switching transient, which fits the requirements of CPU/CPU and CPU/memory interconnections as well as intra-data center connections [3]. Mach-Zehnder interferometer (MZI) is a critical building block for scalable silicon photonic systems, whether in optical

switching or in other applications like optical sensing [4]. The non-resonant interference mechanism for such devices is suitable for spectral broadband operation and temperature-insensitive switching. Many promising achievements have been performed, $32 \times 32$ electro-optic (EO) and $64 \times 64$ thermo-optic (TO) Mach-Zehnder-based switches were fabricated by CAS [5]. To the best of our knowledge, this is the largest scale realized for these types of switches. A $4 \times 4$ EO + TO Mach-Zehnder-based switch monolithically integrated in a CMOS logic process was demonstrated for the first time by IBM [6]. However, the scalability of switches is still limited mainly by the optical path loss and cumulative crosstalk [7]. An appropriate choice of architecture, which impacts strongly on the overall performance of the switch, can reduce these limitations [8]. In several publications, the comparison of architectures takes into account only the effect of MZIs [3,9] or is mainly based on the calculation of the total insertion loss, without considering any crosstalk [10]. These approaches may not be sufficient to estimate which architecture matrix is the most appropriate for a given technology. In this paper, and for the first time to our knowledge, the two basic performance parameters—optical crosstalk and insertion loss—of both MZIs and waveguide crossings, are considered in one single metric to select the most suitable architecture. Then, after defining the signal-to-crosstalk ratio expression, we show that this metric can be used as a tool for architecture evaluation and comparison. This paper is organized as follows. In the next section, the performance of the switch fabric building blocks is defined. In the third section, we describe four architectures very popular in literature. In the fourth section, we introduce our analytical model of SCR, and we detail the calculus necessary to apply it to the four architectures. This is followed, in the fifth section, by a comparison between architectures, using the proposed SCR metric, when scaling up the numbers of inputs/outputs. The last section presents our Si-based $2 \times 2$ MZI and waveguide crossing characterization results. Then, we conclude.

## 2. Performance Limitations of LS-MZS

This section will discuss the two main performance parameters that should be optimized to commercialize large-scale switch fabrics: insertion loss and optical crosstalk. We will model these limitations for MZI and waveguide crossing, which will also be useful to introduce our formalism later.

### 2.1. Insertion Loss

Insertion loss is a major challenge that limits the scaling up of the switch fabric. Losses can reach up to 2 dB per MZI and 0.2 dB per waveguide crossing. Unfortunately, these two components are needed in high numbers to design any switch matrix. Therefore, the architecture chosen must limit the number of cascaded MZIs and waveguide crossings. The power attenuation can be modeled by a loss factor $l_{MZI}$ and $l_X$ in the case of an MZI and a waveguide crossing, respectively.

$$P_{out_{MZI}} = P_{in} l_{MZI} \tag{1}$$

$$P_{out_X} = P_{in} l_X \tag{2}$$

where $P_{in}$ is the optical input power, $P_{out_{MZI}}$ and $P_{out_X}$ are the output powers of each component, respectively (Figure 1). Insertion loss is then defined as the loss factor in dB, noted $Loss_{MZI}$ and $Loss_X$ in the following.

**Figure 1.** Unwanted powers are denoted in blue, $m$ and $x$ represent, respectively, the factors of the leaked power portions in the MZI (**a**) and (**b**), and waveguide crossing (**c**).

## 2.2. Optical Crosstalk

Optical crosstalk occurs when a portion of the signal power leaks into an unwanted output, which could be generated in both MZI and waveguide crossing because of their design. We model these leakages by a factor $m$ in MZI and $x$ in waveguide crossing (see Figure 1). The values of $m$ and $x$ are usually less than 0.01. Crosstalk ratio is defined as the difference, in logarithmic scale, of the output power on the destructive port to the one on the constructive port. The crosstalk issue in MZI is largely due to phase errors or power imbalance inside the arms of the Mach-Zehnder. The phase error can be corrected by introducing heaters on both sides [11], whereas the power imbalance is produced by intrinsic absorptions in waveguide when using electro-optic phase-shifting diode junctions [12] and by the imbalance in 3-dB couplers. Optical crosstalk ratio in MZI and waveguide crossing, named $Xtalk_{MZI}$ and $Xtalk_X$, can be written as follows:

$$Xtalk_{MZI} = 10log_{10}\left(\frac{mP_{in}l_{MZI}}{(1-m)P_{in}l_{MZI}}\right) \approx 10log_{10}(m) \tag{3}$$

$$Xtalk_X = 10log_{10}\left(\frac{xP_{in}l_X}{(1-x)P_{in}l_X}\right) \approx 10log_{10}(x) \tag{4}$$

In the rest of the paper, the unwanted power, written in blue in Figure 1, will be called crosstalk power. It is worth mentioning that there are generally two types of crosstalk, leakage from other signal sources and multi-path interference of the original signal. In this work, they will be treated as purely additive crosstalk.

## 3. Large-Scale Photonic Switch Architectures

We review four architectures–Beneš, dilated Beneš, double layer network, switch and select—that are widely discussed in literature. In the following, we will use the words Stage to refer to a stack of MZIs in the same line, and interconnection to describe the set of waveguides connecting two stages.

The Beneš network [13] is a recursive architecture. To construct $N \times N-$Beneš, two blocs of $N/2 \times N/2-$Beneš are put in the center, and themselves connected to the input/output stages with the interconnection shown in Figure 2. Although it has the smallest overall number of MZIs and stages amongst the four different architectures studied in this paper, it generates first-order crosstalk because each MZI is crossed by two signals, drastically decreasing the overall performance.

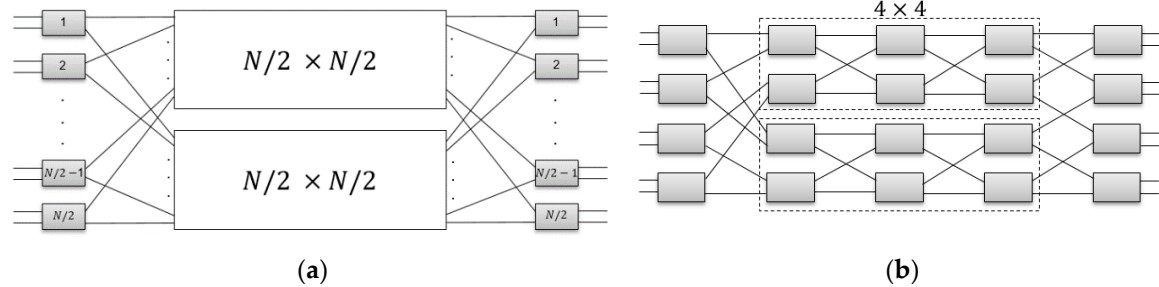

**Figure 2.** (**a**) $N \times N$–Beneš, (**b**) $8 \times 8$–Beneš.

In order to mitigate the crosstalk in Beneš, Padmanabhan and Netravali have proposed a dilated Beneš topology [14]. The recursion of this architecture is similar to the Beneš one, but only one signal travels through any MZI (Figure 3), so that the first-order crosstalk can be fully avoided.

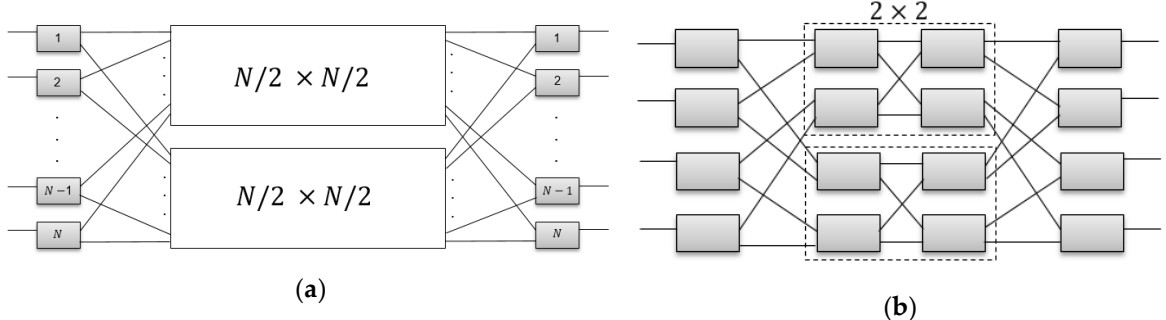

**Figure 3.** (**a**) $N \times N$–Dilated Beneš, (**b**) $4 \times 4$–Dilated Beneš.

The double-layer network (DLN) was proposed by Lu and Thompson [15]. A $N \times N$–DLN consists of two layers, each one composed of two $N/2 \times N/2$–DLN and connected to the input/output stages with the interconnection shown in Figure 4. The number of stages is as low as in Beneš, but the total number of MZIs increases rapidly for a large value of $N$.

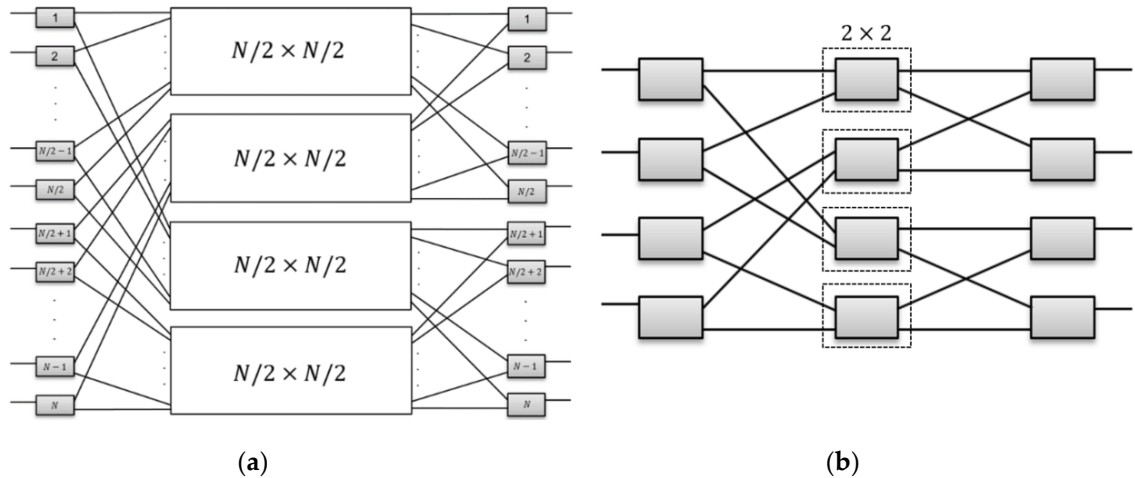

**Figure 4.** (**a**) $N \times N$–DLN, (**b**) $4 \times 4$–DLN.

The switch and select (S&S) consists of $1 \times N$ and $N \times 1$ switch units, shown in Figure 5 [16]. To build a $N \times N$ structure, $N$ $(1 \times N)$ and $N$ $(N \times 1)$ blocks are used and connected via a central interconnection. This topology has the lowest crosstalk caused by MZIs. Nevertheless, the calculation

of the number of waveguide crossings exhibits quadratic growth. Table 1 shows the number of MZIs and waveguide crossings we find for the worst path, and for each topology.

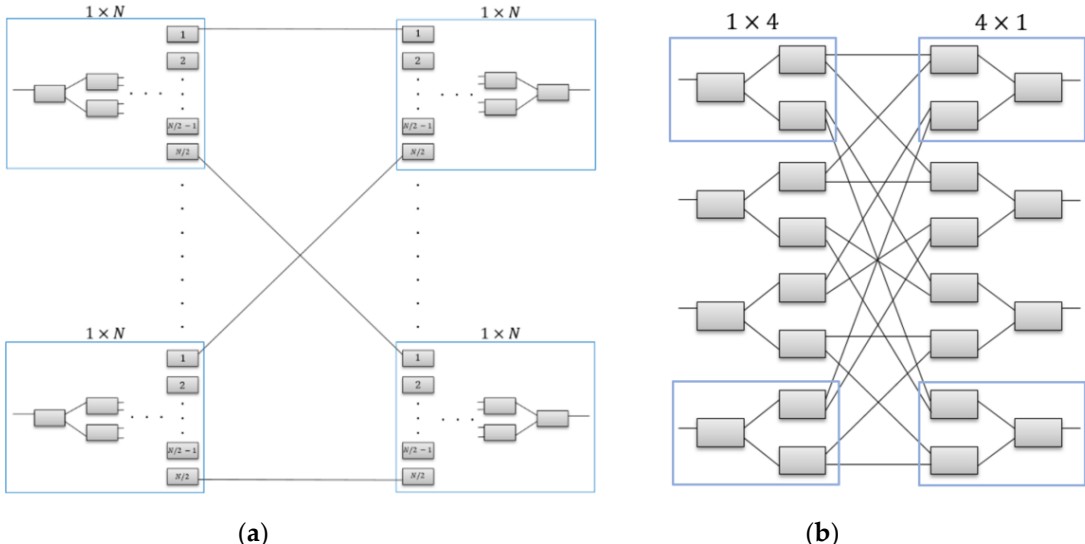

**Figure 5.** (**a**) $N \times N$–S&S, (**b**) $4 \times 4$–S&S.

**Table 1.** Topology comparison for the first case in terms of stage and waveguide crossing counts $N$ is the number of inputs/outputs.

|  | **Stage Count** | **W. Crossing Count** |
|---|---|---|
| Beneš | $2log_2(N) - 1$ | $2(N - log_2(N) - 1)$ |
| Dilated Beneš | $2log_2(N)$ | $2(2N - log_2(N) - 3)^*$ |
| DLN | $2log_2(N) - 1$ | $3N - 2log_2(N) - 4$ |
| S&S | $2log_2(N)$ | $(N - 1)^2$ |

\* Valid for $N > 2$, for $N = 2$: Crossing count = 1.

## 4. Analytical Model: Signal-to-Crosstalk Ratio

In this section, we introduce an analytical model of the SCR of a switch network. Then, we show how to apply it to Beneš, dilated Beneš, switch and select, and double-layer network.

### 4.1. Signal-to-Crosstalk Ratio Model

We start by calculating the SCR of a simple example: a $4 \times 4$–Beneš. Then, we propose a general equation that can be adapted to any architecture. The $4 \times 4$–Beneš structure is shown in Figure 6. The four inputs $P_{in1}$, $P_{in2}$, $P_{in3}$, $P_{in4}$ are connected to the four outputs through the appropriate colored paths. In this example, we have chosen the worst combination connection among the 4! possible ones. We are working on the assumption that the first order crosstalk caused by MZI ($m$) and waveguide crossing ($x$) is not negligible, and so is the second-order crosstalk generated by MZI ($m^2$). The other contributions are neglected: for instance, optical crosstalk caused by an MZI could be the origin of another optical crosstalk generated at a waveguide crossing, or vice versa. This case is not considered here because it has a small influence on the total SCR. Table 2 details the output power following each stage of the structure in Figure 6. The wanted-signal is written in black, crosstalk generated by MZI and waveguide crossing are respectively mentioned in blue and red. Assuming that all signals are coherent and in-phase, we can write for the input signals $P = P_{in1} = P_{in2} = P_{in3} = P_{in4}$. The SCR at the worst output (out2 or out3) is then:

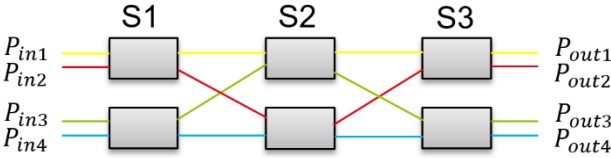

**Figure 6.** $4 \times 4$–Beneš, all MZIs are in bar-state.

$$10log\left(\frac{Pl_{MZI}{}^3l_X{}^2}{mPl_{MZI}{}^3(l_X{}^2 + l_X + 1) + m^2Pl_{MZI}{}^3(l_X + l_X + 1) + xPl_{MZI}{}^3(l_X{}^2 + l_X{}^2)}\right) \tag{5}$$

which simplifies to:

$$10log\left(\frac{l_X{}^2}{m(l_X{}^2 + l_X + 1) + m^2(2l_X + 1) + 2xl_X{}^2}\right) \tag{6}$$

**Table 2.** Signal and cumulative crosstalk generated by MZIs and by waveguide crossings after each stage, for a $4 \times 4$–Beneš when all MZIs are in bare-state. The wanted-signal is written in black, while crosstalk generated by MZI and waveguide crossing are respectively written in blue and red.

| Input and Path Number | After S1 | After S2 | After S3 |
|---|---|---|---|
| $P_1$ | $P_1l_{MZI}$ $+mP_2l_{MZI}.$ | $P_1l_{MZI}{}^2$ $+mP_2l_{MZI}{}^2 +$ $mP_3l_{MZI}{}^2l_X +$ $m^2P_4l_{MZI}{}^2l_X.$ | $P_1l_{MZI}{}^3$ $+mP_2l_{MZI}{}^3 + mP_3l_{MZI}{}^3l_X + m^2P_4l_{MZI}{}^3l_X$ $+mP_2l_{MZI}{}^3l_X{}^2 + m^2P_1l_{MZI}{}^3l_X{}^2 + m^2P_4l_{MZI}{}^3l_X.$ |
| $P_2$ | $P_2l_{MZI}$ $+mP_1l_{MZI}.$ | $P_3l_{MZI}{}^2l_X$ $+mP_4l_{MZI}{}^2l_X +$ $mP_1l_{MZI}{}^2 + m^2P_2l_{MZI}{}^2$ $+xP_2l_{MZI}{}^2l_X.$ | $P_2l_{MZI}{}^3l_X{}^2$ $+mP_1l_{MZI}{}^3l_X{}^2 + mP_4l_{MZI}{}^3l_X + m^2P_3l_{MZI}{}^3l_X$ $+mP_1l_{MZI}{}^3 + m^2P_2l_{MZI}{}^3 + m^2P_3l_{MZI}{}^3l_X$ $+xP_3l_{MZI}{}^3l_X{}^2 + xP_3l_{MZI}{}^3l_X{}^2.$ |
| $P_3$ | $P_3l_{MZI}$ $+mP_4l_{MZI}$ | $P_2l_{MZI}{}^2l_X$ $+mP_1l_{MZI}{}^2l_X +$ $mP_4l_{MZI}{}^2 + m^2P_3l_{MZI}{}^2$ $+xP_3l_{MZI}{}^2l_X.$ | $P_3l_{MZI}{}^3l_X{}^2$ $+mP_4l_{MZI}{}^3l_X{}^2 + mP_1l_{MZI}{}^3l_X + m^2P_2l_{MZI}{}^3l_X$ $+mP_4l_{MZI}{}^3 + m^2P_3l_{MZI}{}^3 + m^2P_2l_{MZI}{}^3l_X$ $+xP_2l_{MZI}{}^3l_X{}^2 + xP_2l_{MZI}{}^3l_X{}^2.$ |
| $P_4$ | $P_4l_{MZI}$ $+mP_3l_{MZI}$ | $P_4l_{MZI}{}^2$ $+mP_3l_{MZI}{}^2 +$ $mP_2l_{MZI}{}^2l_X +$ $m^2P_1l_{MZI}{}^2l_X.$ | $P_4l_{MZI}{}^3$ $+mP_3l_{MZI}{}^3 + mP_2l_{MZI}{}^3l_X + m^2P_1l_{MZI}{}^3l_X$ $+mP_3l_{MZI}{}^3l_X{}^2 + m^2P_4l_{MZI}{}^3l_X{}^2 + m^2P_1l_{MZI}{}^3l_X.$ |

Note that if we neglect the effect of waveguide crossings, i.e., $x \approx 0$, $l_X \approx 1$ and crosstalk of second-order $m^2 \approx 0$, SCR is equal to $-10log(3m)$, which is the equation obtained in [15]. Based on the example of $4 \times 4$−Beneš structure, we notice that the wanted-signal travels through 3 MZIs and 2 waveguide crossings, which represents the worst path. It can also be noted that crosstalk powers at a given output come from different paths that have the same number of stages but a different number of waveguide crossings, so the total insertion loss for each crosstalk power is not always the same. In a large-scale network, it will be difficult to calculate the exact path of each crosstalk power, thus, we assume for simplification purposes that all crosstalk powers pass through an average number of waveguide crossings. Conversely, the number of stages is always the same for all crosstalk powers. All parameters we have taken into consideration in our model are included in Equation (7).

$$S\hat{C}R = 10log\left(\frac{P.l_{MZI}{}^{num_{MZI}}l_X{}^{num_X}}{K_{MZI}(N, m)Pl_{MZI}{}^{num_{MZI}}l_X{}^{M_X} + K_X(N, x)Pl_{MZI}{}^{num_{MZI}}l_X{}^{M_X}}\right) \tag{7}$$

We note $num_{MZI}$ and $num_X$ respectively the number of stages and waveguide crossings in the worst path for a given architecture (Table 1). $K_{MZI}(N,m)P$ and $K_X(N,x)P$ are, respectively, the total cumulative crosstalk powers at the worst-output generated by MZIs and waveguide crossings. The wanted-signal is weighted in the expression by $l_{MZI}{}^{num_{MZI}}l_X{}^{num_X}$, which corresponds to the worst path, while the cumulative crosstalk powers are weighted by $l_{MZI}{}^{num_{MZI}}l_X{}^{M_X}$. $M_X$ is the average of the number of waveguide crossings in the worst and the best paths. We add a hat operator on SCR to highlight an approximation. $N$ is the size of the topology. The equation can be simplified as follows:

$$S\hat{C}R = 10log\left(\frac{l_X{}^{num_X - M_X}}{K_{MZI}(N,m) + K_X(N,x)}\right) \tag{8}$$

The loss factor $l_{MZI}{}^{num_{MZI}}$ is canceled out as a common factor in the numerator and denominator. In the example of $4 \times 4$–Beneš: $K_{MZI}(4,m) = 3m + 3m^2$, $K_X(4,x) = 2x$ and $M_X = \frac{num_X}{2} = 1$ because the number of waveguide crossings equals two in the worst path and zero in the best path, hence,

$$S\hat{C}R = 10log\left(\frac{l_X}{3m + 3m^2 + 2x}\right) \tag{9}$$

It is important to note that Equation (9) is equivalent to Equation (6), considering that the factor $l_X$ is very close to 1 (waveguide crossing losses varying between 0.03 dB and 0.21 dB in the literature). We should keep in mind that our model is based on two main assumptions. The first one is that all crosstalk powers pass through the same number of MZIs, which is always the case in the multi-stage architectures studied. The second one is that all crosstalk powers also pass through the same number of waveguide crossings ($=M_X$), which enables us to keep the calculus simple. The latter assumption could decrease the accuracy of our model, especially when crossing losses are significant. However, we will see in Appendix A that our model remains valid even with relatively high crossing losses, 0.2 dB. Now, our general analytical model is completely defined by Equation (8), we can then apply it to Beneš, dilated Beneš, switch and select, and double-layer network. To do so, the $num_X - M_X$ exponent that we need for each architecture can be easily calculated from Table 1. All that remains is to compute the terms $K_{MZI}(N,m)$ and $K_X(N,x)$, what we are going to do in the next subsection. For simplification purposes, the terms $K_{MZI}(N,m)$ and $K_X(N,x)$ will be simply called cumulative crosstalk powers generated by MZIs and waveguide crossings, respectively.

*4.2. Cumulative Crosstalk Power Terms*

We first calculate the cumulative crosstalk power generated by MZIs. As seen before, for $4 \times 4$–Beneš: $K_{MZI}(4,m) = 3m + 3m^2$. Beneš is a recursive structure, thus, it is not difficult to determine $K_{MZI}(N,m)$ for a higher scale. For $8 \times 8$–Beneš, we find $5m + 10m^2$. Then, we can derive a general formula for $N \times N$–Beneš, with $N = 2^n$ where $\in \mathbb{N}^*$:

$$K_{MZI}(N,m) = (2k-1)m + \left(2k^2 - 3k + 1\right)m^2 \tag{10}$$

We note $k = log_2(N)$. The results for the other three architectures are mentioned in [15,17] and summarized in Table 3. We still have to calculate cumulative crosstalk power from waveguide crossings. In the Beneš matrix, all waveguides are crossed by optical power, so all crossings are contributors to crosstalk. In the worst case:

$$K_X(N,x) = 2(N - k - 1)x \tag{11}$$

For dilated Beneš (Figure 3), only one signal passes through each MZI, and therefore not all waveguide crossings are crossed by signals for a given network configuration. The cumulative crosstalk power induced by waveguide crossings in the worst case is:

$$K_X(4,x) = 4x \tag{12}$$

$$K_X(8, x) = \frac{8}{2}x + K_X(4, x) + \frac{8}{2}x \tag{13}$$

$$K_X(N, x) = N/2x + K_X(N/2, x) + N/2x \tag{14}$$

By adding up:

$$K_X(N, x) = 2(N - 2)x \tag{15}$$

Similarly, for DLN (Figure 4):

$$K_X(4, x) = \frac{4}{2}x + x \tag{16}$$

$$K_X(8, x) = \frac{8}{2}x + K_X(4, x) + \frac{8}{4}x \tag{17}$$

$$K_X(N, x) = N/2x + K_X(N/2, x) + N/4x \tag{18}$$

We get:

$$K_X(N, x) = 3(N/2 - 1)x \tag{19}$$

For $N \times N$–S&S (Figure 5), each $1 \times N$–bloc is crossed by only one signal. Therefore, it is not difficult to find:

$$K_X(N, x) = (N - 1)x \tag{20}$$

The following table summarizes the values corresponding to each topology. Note that an architecture's total crosstalk power could be slightly reduced by introducing intelligent algorithms [18].

**Table 3.** Topology comparison in terms of cumulative crosstalk generated by MZIs and waveguide crossings for each topology.

| | $K_{MZI}(N,m)$ | $K_X(N,x)$ |
|---|---|---|
| Beneš | $(2k - 1)m + \left(2k^2 - 3k + 1\right)m^2$ | $2(N - k - 1)x$ |
| Dilated Beneš | $k(2k - 1)m^2$ | $2(N - 2)x^*$ |
| DLN | $m + (k - 1)m^2$ | $3(N/2 - 1)x$ |
| S&S | $km^2$ | $(N - 1)x$ |

* Valid for $N > 2$, for $N = 2$: $K_X(2, x) = x$.

All the expressions needed are now computed for the four architectures. We will depict $S\hat{C}R$ values as a function of $N$ for the four architectures in order to compare them; this will be the subject of the next section.

## 5. Architecture Comparison

In the following section, we calculate $S\hat{C}R$ of the four architectures based on the component performances found in the literature. Today's best results are presented in Table 4.

Comparison between the four topologies shows a significant impact of waveguide crossings on the overall performance of the network. S&S is the best to reduce first-order and second-order crosstalk powers generated by MZIs. Nevertheless, it does not scale well because its central interconnection contains a high number of waveguide crossings, as shown in Figure 7a. The waveguide crossing count in the dilated Beneš is also important; this has a strong impact when its insertion loss is not optimized. Indeed, in Figure 7b and for $Loss_X = 0.21$ dB [19], the dilated Beneš $S\hat{C}R$ (solid red line) decreases more rapidly and becomes lower than Beneš $S\hat{C}R$ for $N = 32$. In general, each architecture can be badly affected if the waveguide crossing performances are not sufficiently optimized, as seen in Figure 7c, with $Xtalk_X = -20$ dB [20]. The four architectures have similar performances with a $S\hat{C}R$ lower than 17 dB for $N \geq 4$. DLN seems to be the most favorable architecture, whose $S\hat{C}R$ value decreases more slowly. Its major challenge is the overall number of MZIs, which is relatively large.

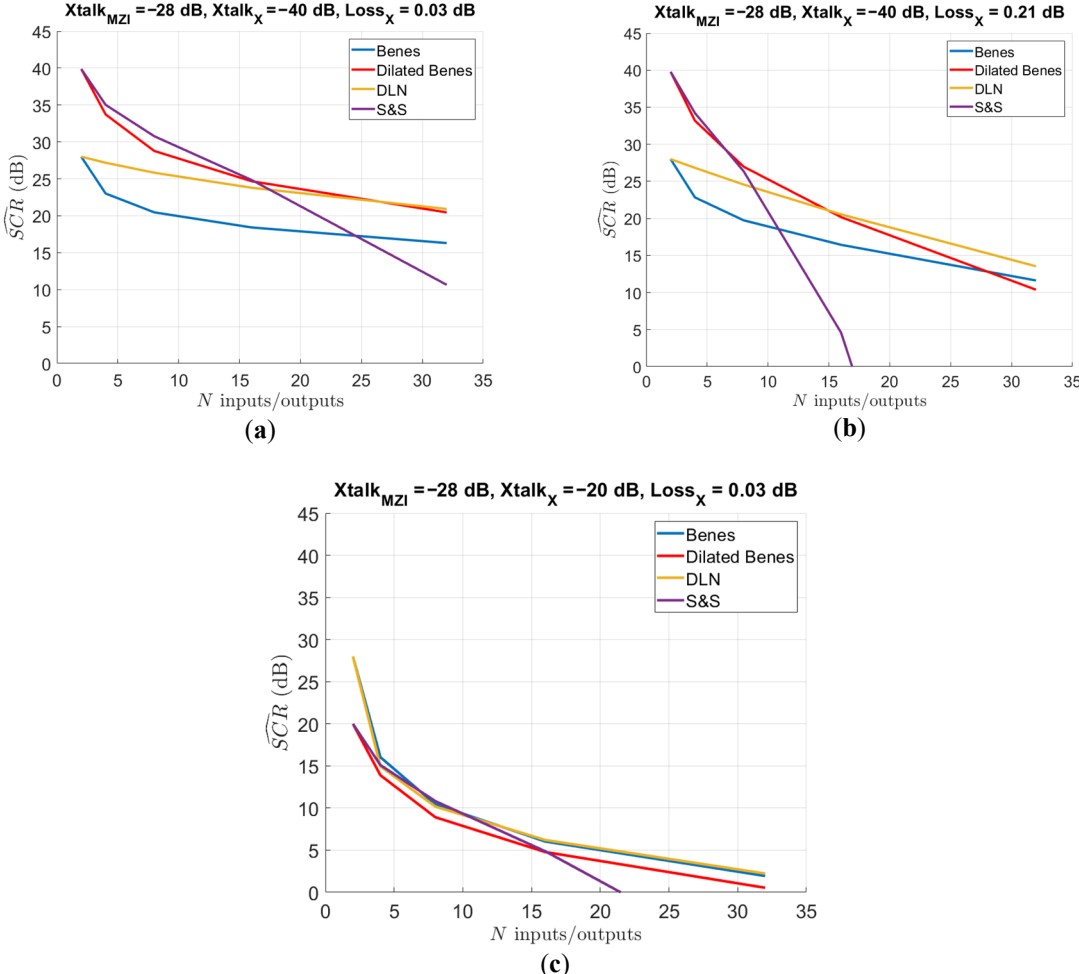

**Figure 7.** $S\hat{C}R$ of the four topologies using the state-of-the-art loss and crossing values (**a**), a significant crossing loss $Loss_X = 0.21$ dB (**b**) and using a non-optimized crossing crosstalk $Xtalk_X = -20$ dB (**c**).

**Table 4.** State-of-the-art loss and crosstalk of MZI and waveguide crossing.

| Metric | Value | Ref. |
| --- | --- | --- |
| Waveguide crossing loss ($Loss_X$) | 0.03 dB | [21] |
| Waveguide crossing crosstalk ($Xtalk_X$) | $<-40$ dB | [21] |
| MZI Loss ($Loss_{MZI}$) | 0.8 dB | [22] |
| MZI Crosstalk ($Xtalk_{MZI}$) | $-28$ dB | [22] |

## 6. MZI and Waveguide Crossing Characteristics

In this section, we present the performances of our MZI and waveguide crossing fabricated in LETI's 200 mm silicon photonics fabrication platform, with which we will plot the $S\hat{C}R$ values of the four topologies.

We designed a 2 × 2 MZI circuit, Figure 8 [23]. The structure has 3-dB multimode interference couplers (M1 and M2), a 250 μm-length PIN diode phase shifter in each arm (PS1 and PS2), a 50 μm-length heater in each arm too (H1 and H2) and a photodetector at each output-side (P1 and P2). A loop-back waveguide was added for fiber alignment and for evaluating the insertion losses of MZI. Measurements of optical power on Out1 and Out2 were performed using a fiber array feeding the input In1. The bar-state and cross-state were observed depending on the change in phase shift. For that, a current of up to 12 mA is first injected into PS1, and then the same protocol was applied to

PS2. The obtained results of the two-phase shifters were combined and depicted in Figure 9a. It is important to point out that our MZI operates using the two electro-optic (EO) phase shifters (PS1 and PS2), whereas the two heaters (H1 and H2) were specifically integrated to correct static phase errors. The results presented below were obtained without phase corrections, that is to say, only EO phase shifters were operated.

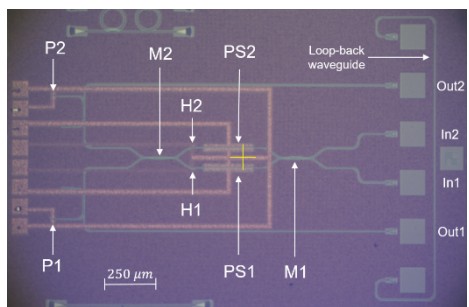

**Figure 8.** Image of 2 × 2 MZI circuit: M1 and M2 are 3-dB MMI couplers; the phase shifters (PS1 and PS2) are commonly grounded and symmetrically integrated; H1 and H2 are the heaters, P1 and P2 are the photodetectors.

For the best die, we measured a crosstalk of −33.4 dB in bar-state, and a crosstalk of −31.3 dB in cross-state. The transmission spectrum in both states and for the loop-back waveguide is depicted in Figure 9b, the insertion loss recorded in the MZI equals 6.42 dB at the best wavelength, but this value includes the grating coupler losses. In the loop-back waveguide, the insertion loss is 5.11 dB. Therefore, the MZI insertion loss is estimated near 1.31 dB. It is important to note that, at the wafer scale, the calculated mean value of crosstalk is −25 dB, and by testing the other input (Out2 instead of Out1), the same value was found. It is noteworthy that these measurements were carried out at room temperature. We did not change the temperature to investigate its effect on the MZI, but we think that it is quite resistant to temperature changes. Waveguide crossing was also characterized; the results obtained are shown in Figure 9c. The measured crosstalk is lower than −42 dB, with an insertion loss of 0.1 dB. Our best components in performances are summarized in Table 5.

**Table 5.** Best component performances.

| Metric | Value |
| --- | --- |
| Waveguide crossing loss ($Loss_X$) | 0.1 dB |
| Waveguide crossing crosstalk ($Xtalk_X$) | <−42 dB |
| MZI Loss ($Loss_{MZI}$) | ~1.31 dB |
| MZI Crosstalk ($Xtalk_{MZI}$) | −31.3 dB |

Compared to Table 4 values, our technology has better performances in terms of crosstalk, but lower performances in terms of loss.

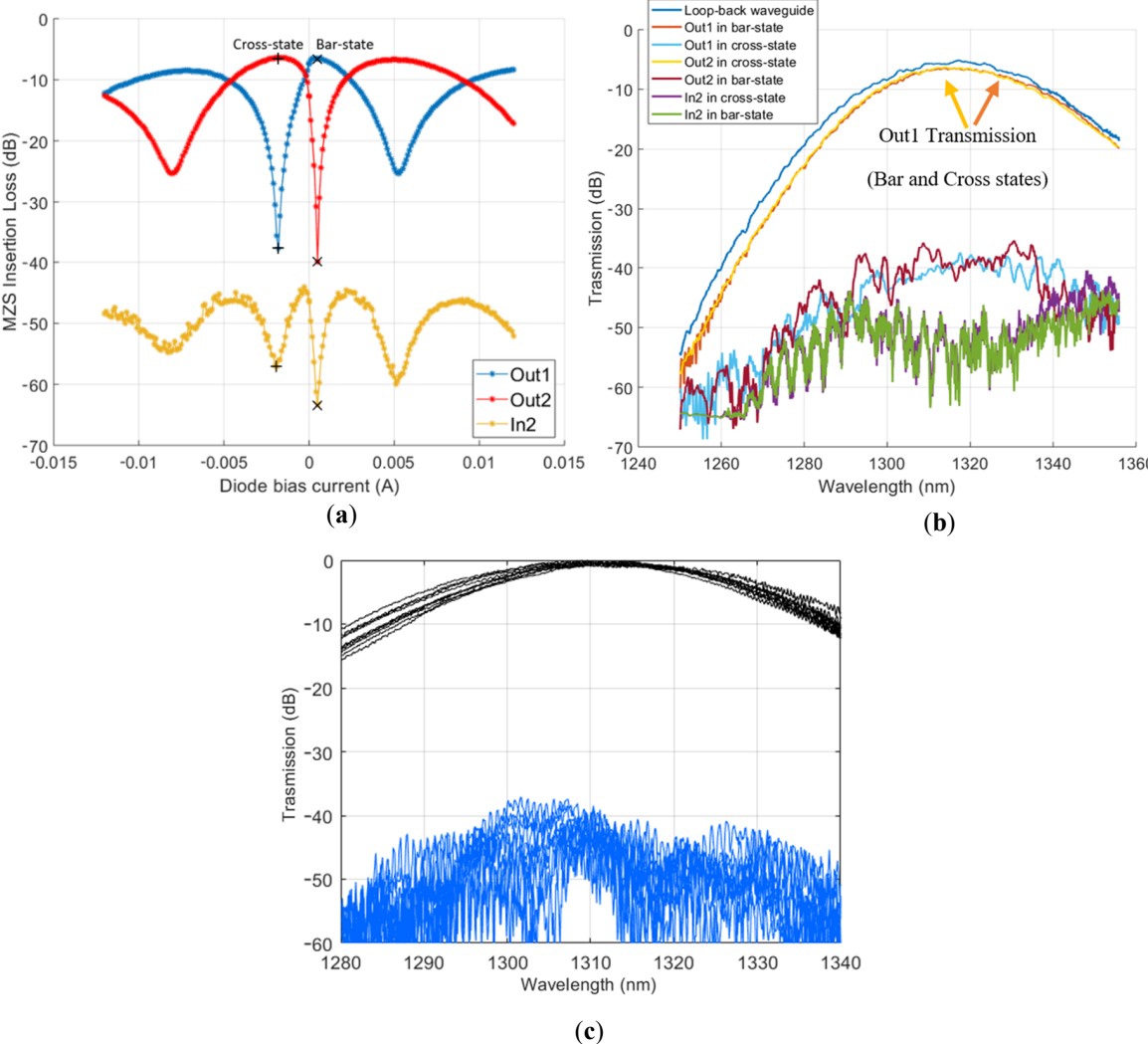

**Figure 9.** (**a**) Insertion loss measured of 2 × 2 MZI by varying the biased current separately in the two-phase shifters. (**b**) Transmission signal spectrum (in dB) of the loop-back waveguide and the MZI in bar and cross-states. (**c**) Measured transmission spectra (in dB) of waveguide crossing (dark lines) and crosstalk recorded (blue lines).

We compared the four topologies discussed before, based on these performances; Figure 10 shows $S\hat{C}R$ values for each of them. DLN scales better than the others did, a 32 × 32 switch can be fabricated with an $S\hat{C}R \geq 20$ dB. However, we can notice that it is badly impacted by the first-order MZI crosstalk generated at the center stage, unlike dilated Beneš and S&S, whose $S\hat{C}R$s are over 40 dB for $N = 4$. $S\hat{C}R$ values of dilated Beneš is quite good on a small scale ($N \leq 16$), but waveguide crossings loss (0.1 dB/crossing) badly affect its performance for the higher scale. Beneš is a good topology because its $S\hat{C}R$ does not drop quickly. Nevertheless, it requires further optimization of first-order MZI crosstalk. The high number of waveguide crossings in S&S is still a real issue; we think it is not adequate for our technology. Other topologies could be analyzed using our SCR analytical model to choose the most suitable.

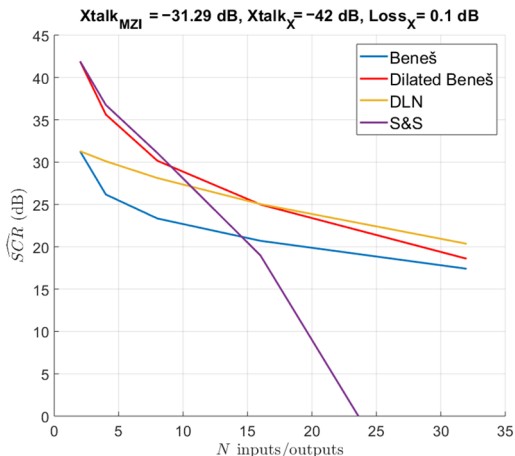

**Figure 10.** $S\hat{C}R$ expected using our technology.

## 7. Discussion

By using our simple metric in order to evaluate and compare between architectures, we found that waveguide crossings are critical components, especially in large-scale switches. Lu and Thompson in [15] and Kabacinski in [17] did not consider the effects of waveguide crossings in their SNR calculations. With these obtained results, we underline its strong influence on the LS-MZS performances. Indeed, S&S is a simple architecture that reduces the crosstalk induced by MZIs, but the central interconnection, composed of a large number of waveguide crossings, considerably limits the overall performance of the network. The high number of waveguide crossings also influences the performance of dilated Beneš. DLN offers the right compromise in terms of stage count, crossing count, and total cumulative crosstalk. Yet, the total number of MZIs increases rapidly. Furthermore, we can also determine the maximum crosstalk and loss values of each component in order to design an LS-MZS of order N.

**Author Contributions:** M.K.: computation of the analytical SCR formula and numerical cross-checks. B.C.: supervision of the computation and interpretation of the results. C.A. direction of the work and supervision of the overall study. All authors have read and agreed to the published version of the manuscript.

**Funding:** This work was supported by the French national program 'programme d'Investissements d'Avenir', IRT Nanoelec ANR-10-AIRT-05.

**Conflicts of Interest:** The authors declare no conflict of interest.

## Appendix A

Here, we show how we simulated Beneš and dilated Beneš numerically in order to test the accuracy of our $S\hat{C}R$ expression. Also, we discuss the results of the simulations.

A switch topology can be represented as a matrix decomposition of multiple stages and interconnections, as shown in Figure A1.

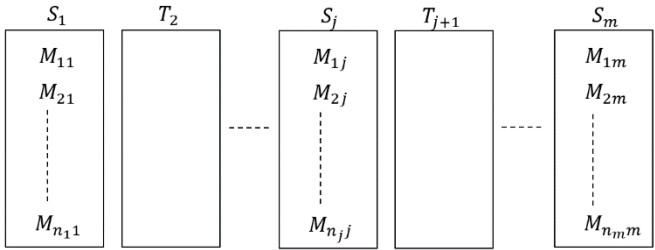

**Figure A1.** Switch network represented by a matrix decomposition of multi-stage of MZIs and interconnections, $S_j$ is a matrix of MZIs in a stage. $T_j$ is the interconnection matrix connecting two stages, $S_{j-1}$ *and* $S_{j+1}$.

With $\left(m,\ n_j\right) \in \mathbb{N}^* \times \mathbb{N}^*$ and $j \in \{1, m\}$. The transfer matrix of the switch network can be seen as a product of the stage matrices:

$$\prod_{j=1}^{m} I_{m-j+1} \tag{A1}$$

With,

$$I_j = \begin{cases} S_j \ if \ j \ an \ odd \ number \\ \qquad T_j \ otherwise \end{cases} \tag{A2}$$

We define,

$$S_j = \begin{bmatrix} M_{1j} \\ M_{2j} \\ . \\ . \\ . \\ M_{n_j j} \end{bmatrix} \tag{A3}$$

For $i \in \left\{1, n_j\right\}$:

$$M_{ij} = \begin{cases} l_{MZI}\begin{pmatrix} 1-m & m \\ m & 1-m \end{pmatrix} \text{in bare state} \\ l_{MZI}\begin{pmatrix} m & 1-m \\ 1-m & m \end{pmatrix} \text{in cross state} \end{cases} \tag{A4}$$

The State of each $M_{ij}$ depends on the network configuration. $T_j$ is the interconnection matrix connecting two following stages, $S_{j-1}$ and $S_{j+1}$. This matrix model is detailed through the example of $4 \times 4 - Benes$ (Figure 6). The following interconnection in Figure A2 can be modeled with the matrix $T$ defined as:

$$T = \begin{pmatrix} 1 & 0 & 0 & 0 \\ 0 & xl_X & (1-x)l_X & 0 \\ 0 & (1-x)l_X & xl_X & 0 \\ 0 & 0 & 0 & 1 \end{pmatrix} \tag{A5}$$

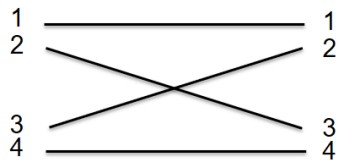

**Figure A2.** Example of an interconnection between 4 MZIs, used in $4 \times 4 - Benes$.

We note: $P_{out1}$, $P_{out2}$, $P_{out3}$, $P_{out4}$, the four powers of the structure outputs. The expression of $4 \times 4 - \text{Benes}$ can therefore be written as follows:

$$\begin{pmatrix} P_{out1} \\ P_{out2} \\ P_{out3} \\ P_{out4} \end{pmatrix} = \begin{pmatrix} M_{15} \\ M_{25} \end{pmatrix} T \begin{pmatrix} M_{13} \\ M_{23} \end{pmatrix} T \begin{pmatrix} M_{11} \\ M_{21} \end{pmatrix} \begin{pmatrix} P_{in1} \\ P_{in2} \\ P_{in3} \\ P_{in4} \end{pmatrix} \tag{A6}$$

and,

$$M_{ij} = l_{MZI} \begin{pmatrix} 1-m & m \\ m & 1-m \end{pmatrix} \text{ for } i, j \, \epsilon \, \{1,2\} \times \{1,3,5\} \tag{A7}$$

We have applied the matrix approach to implement Beneš and dilated Beneš for 2, 4, 8, and 16 switch sizes. Then we have tested several permutations of MZIs, corresponding to various possible paths, with the consideration that only one signal passes through an MZI in dilated Beneš. The performance parameters used are: $Xtalk_{MZI} = -20$ dB, $Xtalk_X = -25$ dB, $Loss_X = 0.03$ dB. The implementation of the matrix models was developed with Python, and the simulations were limited to N ≤ 16 because of the time requirement. The matrix simulation results, in black square dots on Figure A3 for both architectures, fit well to our $S\hat{C}R$ model ones plotted as the solid blue line. In order to investigate the waveguide crossings effects, Figure A3a traces the $S\hat{C}R$ values of Benes without taking into account the effects of waveguide crossings (solid red line), i.e., $x = 0$ and $l_x = 1$. As we can notice, the difference between the two curves increases with the number of inputs/outputs; this is due to the number of waveguide crossings becoming high in a large network. For dilated Beneš, the results are presented in Figure A3b. We observe that the gap between the two curves is even higher (from 15 to 18 dB to be compared to 5 dB max for Beneš), that is because the waveguide crossing count in this architecture is much higher.

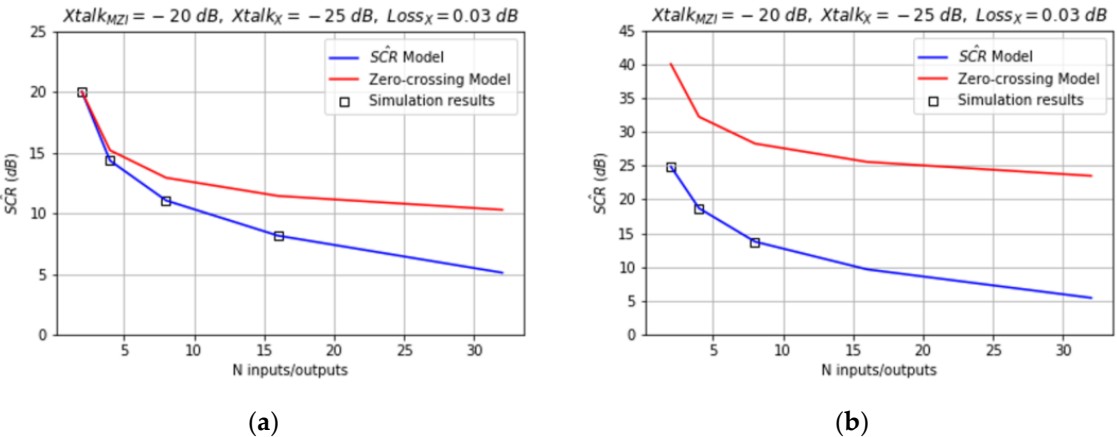

(a)          (b)

**Figure A3.** $S\hat{C}R$ values of Benes (**a**) and dilated Benes (**b**): solid blue curves for the complete model; solid red curves for the zero-crossing model. Black square dots are the simulation results.

As mentioned in Section 4.2, the accuracy of our model could decrease when waveguide crossing losses are important. For verification purposes, we simulated Beneš considering $Loss_X = 0.2$ dB, relatively high losses. The results are shown in Figure A4. We can see that our $S\hat{C}R$ model still gives a good approximation (less than 1.5 dB difference between $S\hat{C}R$ value and simulation result in the case of Beneš of order 16). This shows that our mathematical model remains valid even if the waveguide crossing losses are significant.

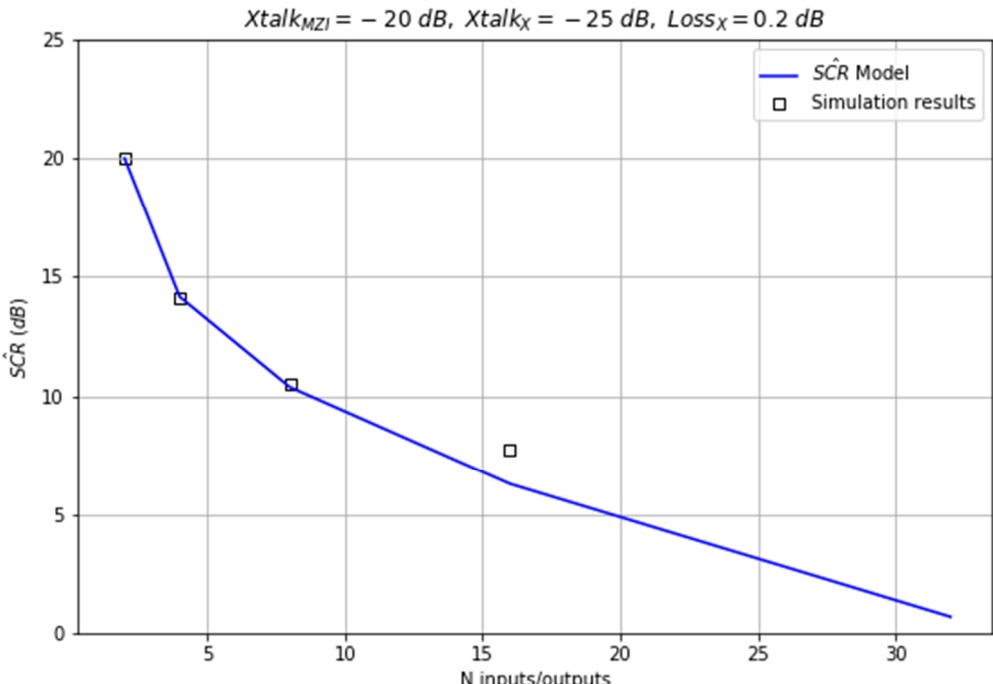

**Figure A4.** *SĈR* values of Bene*s* considering *Loss$_X$* = 0.2 *dB*, the solid blue curve. Black square dots are the simulation results.

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
