# Peer review of "Comprehensive Model for Evaluating the Performance of Mach-Zehnder-Based Silicon Photonic Switch Fabrics in Large Scale"

_applsci, doi:10.3390/app10238688_

Round 1

Reviewer 1 Report

This manuscript describes estimation of optical signal-to-crosstalk ratios and insertion losses of silicon photonic switch circuits of 4 different topological structures for large scale expansion cases based on their measured device parameters of Mach-Zehnder interferometer and waveguide crossing. The reported results will be useful to experts in this technical area. I would recommend that this manuscript could be accepted for publication in Applied Sciences after some minor improvements on the following comments:

- In abstract, it is mentioned that the 2x2 MZI has two electro-optic (EO) phase shifters. However, in Fig. 8 and corresponding text description, two heaters were used for the phase shifters which are probably thermo-optic (TO) ones. Authors should clarify this matter. It is also recommended to add some comments on the electrical power consumption for the TO case or on the switching speed for the EO case.

- Table 5 shows the best performance data of the fabricated components. It is curious how the device performance varies from chip to chip. Authors may add some comments on this matter.

End

Author Response

We would like to thank you for your careful reading of our manuscript, for your supportive comments and for your thoughtful remarks, which will certainly help us to improve this manuscript. The changes made in the revised paper are summarized below.

Your observations about phase shifters are exact. In fact, two PIN diodes (one in each arm) were integrated for dynamic phase shifting, and two heaters (one in each arm) for static phase error corrections, as well. However, the results presented in this paper (section 6) were obtained by operating only the PIN junctions. The following paragraph has been introduced to clarify this point (lines 303-307):

It is important to point out that our MZI operates using the two electro-optic (EO) phase shifters (PS1 and PS2), whereas the two heaters (H1 and H2) were specifically integrated to correct static phase errors. The results presented below were obtained without phase corrections, that is to say, only EO phase shifters were operated.

You are absolutely right, we did not add any comments on switching time. That is because we have already published a conference paper dedicated to the measurement results of this MZI, in which we gave details on this point. The paper reference has been added in line 296 to reflect this lack of information.

It is indeed true that the performance of our components varies from one die to another. For instance, the measured crosstalk of the MZI ranged from -21 dB to -31 dB on 46 dies tested, with an average of -25 dB, as mentioned in line 313 in the paper. In the table 5, only best component performances were reported because our main objective was to apply the SCR formula using our technology and show how this mathematical model could be used to theoretically estimate the performance of large-scale Mach-Zehnder switches depending on the selected architecture. The title of the table 5 has been changed to “Best component performances” in order to elucidate this matter.

Reviewer 2 Report

The paper seems very interesting and the structure is well organized. Please consider the following revisions.

a) Define MZI in the Introduction.

b) I miss some recent references about MZI for general areas. Please consider: 1) https://doi.org/10.3390/s20092640; 2) 10.1109/JSEN.2020.2974931; 3) 10.1016/j.optlastec.2019.105743; among others.

c) Please add some words about the temperature influence in results from Fig. 9.

d) Fig. 10 : I miss associated errors in discussion and graphs.

e) Please add a table for comparison with the proposed work and the literature ones.

Author Response

We would like to thank you for your careful reading of our manuscript, for your supportive comments and for your thoughtful remarks, which will certainly help us to improve this manuscript. The changes made in the revised paper are summarized below.

Indeed, we forgot to define MZI in the introduction, thank you for the reminder. We appreciate the recent references you have suggested to us. The first one is a really interesting paper on MZI sensors. However, we are afraid that the two others are not quite adequate with the topic of this manuscript. Perhaps you were mistaken about these references. The following paragraph (lines 41-45) has been added to address the first and second points, a) MZI definition and b) lack of reference.

Mach-Zehnder Interferometer (MZI) is a critical building block for scalable silicon photonic systems, whether in optical switching or in other applications like optical sensing [4]. The non-resonant interference mechanism to such devices is suitable for spectral broadband operation and temperature-insensitive switching.

c) We would like to thank you for pointing out the temperature sensitive matter. The following paragraph (lines 314-316) has been introduced to reflect this point:

It is noteworthy that these measurements were carried out at room temperature. We did not change the temperature to investigate its effect on the MZI, but we think that it is quite resistant to temperature changes.

d) The results of Fig. 10 are theoretical values of the SCR because they are calculated using our mathematical model. It is sure that these values are not perfectly accurate compared to the numerical ones, this issue is discussed in appendix A and shown in the results of graph A3 and A4. In the paragraph above Fig. 10 (lines 341-350), we tried to discuss these results and especially to compare the architectures in large-scale to see the most adequate to our technology.

e) We fully agree that adding a table to compare our work with literature ones could be a didactic way, but from our perspective, the discussion (section 7) emphasizes that our proposed work takes into account the effects of waveguide crossings, unlike the work of Lu and Thompson and that of Kabacinski. Therefore, we think that adding one more table could be redundant.

Round 2

Reviewer 2 Report

Point b of my comment is not fully addressed. Please consider to include the suggestion.